# Selection of Gut-Resistant Bacteria and Construction of Microbial Consortia for Improving Gluten Digestion under Simulated Gastrointestinal Conditions

**DOI:** 10.3390/nu13030992

**Published:** 2021-03-19

**Authors:** Maria De Angelis, Sonya Siragusa, Mirco Vacca, Raffaella Di Cagno, Fernanda Cristofori, Michael Schwarm, Stefan Pelzer, Monika Flügel, Bodo Speckmann, Ruggiero Francavilla, Marco Gobbetti

**Affiliations:** 1Department of Soil, Plant and Food Sciences, University of Bari Aldo Moro, 70126 Bari, Italy; maria.deangelis@uniba.it (M.D.A.); sonya.siragusa@uniba.it (S.S.); mirco.vacca@uniba.it (M.V.); 2Faculty of Science and Technology, Free University of Bozen, 39100 Bolzano, Italy; raffaella.dicagno@unibz.it; 3Interdisciplinary Department of Medicine-Pediatric Section, University of Bari Aldo Moro, Ospedale Pediatrico Giovanni XXIII, 70125 Bari, Italy; fernandacristofori@gmail.com (F.C.); ruggiero.francavilla@uniba.it (R.F.); 4Evonik Operations GmbH, 63457 Hanau-Wolfgang, Germany; michael.schwarm@evonik.com (M.S.); stefan.pelzer@evonik.com (S.P.); monika.fluegel@evonik.com (M.F.); bodo.speckmann@evonik.com (B.S.)

**Keywords:** gluten epitopes, bacterial peptidases, *Bacillus*, *Lactiplantibacillus*, *Lacticaseibacillus*, *Limosilactobacillus*

## Abstract

This work aimed to define the microbial consortia that are able to digest gluten into non-toxic and non-immunogenic peptides in the human gastrointestinal tract. Methods: 131 out of 504 tested *Bacillus* and lactic acid bacteria, specifically *Bacillus* (64), lactobacilli (63), *Pediococcus* (1), and *Weissella* (3), showed strong gastrointestinal resistance and were selected for their PepN, PepI, PepX, PepO, and PepP activities toward synthetic substrates. Based on multivariate analysis, 24 strains were clearly distinct from the other tested strains based on having the highest enzymatic activities. As estimated by RP-HPLC and nano-ESI–MS/MS, 6 cytoplasmic extracts out of 24 selected strains showed the ability to hydrolyze immunogenic epitopes, specifically 57–68 of α9-gliadin, 62–75 of A-gliadin, 134–153 of γ-gliadin, and 57–89 (33-mer) of α2-gliadin. Live and lysed cells of selected strains were combined into different microbial consortia for hydrolyzing gluten under gastrointestinal conditions. Commercial proteolytic enzymes (*Aspergillus*
*oryzae* E1, *Aspergillus*
*niger* E2, *Bacillus*
*subtilis* Veron HPP, and Veron PS proteases) were also added to each microbial consortium. Consortium activity was evaluated by ELISA tests, RP-HPLC-nano-ESI–MS/MS, and duodenal explants from celiac disease patients. Results: two microbial consortia (Consortium 4: *Lactiplantibacillus* (*Lp*.) *plantarum* DSM33363 and DSM33364, *Lacticaseibacillus* (*Lc*.) *paracasei* DSM33373, *Bacillus*
*subtilis* DSM33298, and *Bacillus*
*pumilus* DSM33301; and Consortium 16: *Lp*. *plantarum* DSM33363 and DSM33364, *Lc*. *paracasei* DSM33373, *Limosilactobacillus*
*reuteri* DSM33374, *Bacillus*
*megaterium* DSM33300, *B.*
*pumilus* DSM33297 and DSM33355), containing commercial enzymes, were able to hydrolyze gluten to non-toxic and non-immunogenic peptides under gastrointestinal conditions. Conclusions: the results of this study provide evidence that selected microbial consortia could potentially improve the digestion of gluten in gluten-sensitive patients by hydrolyzing the immunogenic peptides during gastrointestinal digestion.

## 1. Introduction

Gluten is a cereal protein network made of high and low molecular weight glutenins (HMW-GSs and LMW-GSs, respectively), S-poor ω-gliadins, and S-rich α-, β-, and γ-gliadins [1,2,3]. While containing small levels of lysine, methionine, threonine, and other amino acids, gliadins are the richest source of glutamine (43%) and proline (13%) in the diet. Because of its unique cyclic structure, proline forms peptide bonds which are particularly resistant to hydrolysis. A portfolio of peptidases (up to nine) is necessary for hydrolyzing all potential gluten polypeptides where proline is located at different positions. Human digestive proteases only partially digest proline-rich gliadins [4,5,6,7]. In general, this incapability limits digestive processes and, in particular, leads to the genesis of gluten-derived peptides (epitopes) which act as triggering factors for food disorders in susceptible individuals [8]. Furthermore, members of the human gut microbiota (e.g., *Neisseria flavescens* and *Pseudomonas aeruginosa*) have the capability to increase the content of these epitopes, which efficiently cross the intestine mucosal barrier [9,10]. 

There is a wide spectrum of gluten-related disorders comprising autoimmunity (celiac disease, CD; food allergy, FA; wheat-dependent, exercise-induced anaphylaxis, WDEIA; respiratory allergy, RA) and innate immunity (non-celiac gluten sensitivity, NCGS) diseases along with gluten-sensitive irritable bowel syndrome (IBS). Despite the different etiologies and varying levels of gluten sensitivity, there is a common therapy for these disorders, namely a gluten-free diet (GFD). Other studies have also shown evidence that GFD alleviates the symptoms of schizophrenia, atopy, fibromyalgia, endometriosis, obesity, and non-specific gastrointestinal symptoms [11]. Compliance with a strict GFD is challenging because of the ubiquitous nature of the protein, food cross-contamination, inadequate food labeling, and social constraints. Even foods claimed to be gluten-free (<20 ppm) may contain traces of gluten, exceeding a safe intake threshold. Moreover, a GFD is often imbalanced, e.g., due to the avoidance of cereal products, causing micronutrient and fiber deficiencies alongside an excess of calories and the increased amount of sugar and saturated fats found in many gluten-replacement foods [12,13]. Long-term adherence to an unbalanced GFD may therefore predispose participants to malnutrition-associated disorders. 

Because of all the medical and dietary implications, the attempts to increase the human capability to digest gluten epitopes deserves marked interest not only in order to eliminate the risks of gluten cross-contamination for CD patients but, also, as a complementary strategy to enable gluten dietary reduction for all other disorders. In general, more efficient gluten degradation might also be beneficial for the health of the population by improving the digestive processes [14]. 

Currently, several microbiota-targeted approaches have been developed with the common aim of ameliorating gluten-related disorders [15,16]. These mainly comprise oral applications of selected bacteria (e.g., lactobacilli and/or *Bifidobacterium* spp.) [17] or protein/peptide hydrolases (e.g., glutenases) [18,19]. Recently, the IBS-like symptoms of CD patients under a GFD were improved after oral administration of a multispecies combination of lactobacilli and *Bifidobacterium* [17]. This treatment is also associated with a beneficial shift in the gut microbiota composition, but concrete evidence for microbial survival and effective gluten degradation under gastrointestinal conditions has not yet been provided. In general, the lack of successful clinical translation of these microbial preparations might be explained by their poor survival under gastrointestinal conditions, the restrictive use of only a few species belonging to lactobacilli and *Bifidobacterium* taxa and, in particular, the scarce activity against gluten epitopes [20,21,22,23]. Another option concerns protein/peptide hydrolases from plants and other microbial sources (e.g., *Bacillus stearothermophilus*, *Bacillus thermoproteolyticus*, *Bacillus licheniformis*, *Streptomyces griseus*, and *Aspergillus niger*) [14,19,24,25]. A major limitation of these formulations is the minimal evidence for their efficacy, and successful clinical translation is lacking due to challenges brought about by the poor resistance to gastrointestinal conditions [25]. Furthermore, these enzyme preparations might also represent a hazard because of the only partial hydrolysis of gluten, which may increase, instead of decrease, the abundance of epitopes [25]. Currently, the scientific consensus addresses novel microbial candidates (e.g., spore-forming species) in combination with the usual lactobacilli and/or *Bifidobacterium* and proteolytic enzymes, which represent the most promising choice, provided that their activity under gastrointestinal conditions is clearly preserved [26]. 

This study aimed at selecting microbial consortia, comprising *Bacillus* sp., lactic acid bacteria, and their cytoplasmic enzymes, added with commercial gluten-hydrolyzing enzymes to prepare mixed formulations capable of digesting gluten into non-toxic and non-immunogenic peptides/amino acids under human gastrointestinal conditions.

## 2. Materials and Methods

### 2.1. Microorganisms and Culture Conditions

This study evaluated 190 strains of the genus *Bacillus* and 314 lactic acid bacteria (LABs). 

In detail, the number of Bacillus assigned at species level were, Bacillus coagulans (n. 1), Bacillus licheniformis (n. 10), Bacillus megaterium (n. 68), Bacillus pumilus (n. 52), Bacillus subtilis (n. 45), Bacillus amyloliquefaciens (n. 1), and Bacillus velezensis (n. 13). 

Thirty-two out of 314 LABs were not assigned at genus level even if they belong to the family of *Lactobacillaceae;* therefore, these were reported as lactobacilli. Instead, *Lactobacillaceae* assigned at least at genus level were: *Levilactobacillus (Lv.) brevis* (n. 42), *Lacticaseibacillus (Lc.) casei* (n. 14), *Limosilactobacillus (Ls.) fermentum* (n. 11), *Lacticaseibacillus paracasei* (n. 22), *Lactiplantibacillus (Lp.) plantarum* (n. 113), *Limosilactobacillus reuteri* (n. 1), *Furfurilactobacillus (F.) rossiae* (n. 14), *Fructilactobacillus (Fr.) sanfranciscensis* (n. 36), *Leuconostoc (Leu.) mesenteroides* (n. 6), *Pediococcus (Ped.) pentosaceus* (n. 5), and *Weissella* sp. (n. 18). 

Strain isolation was achieved via food matrixes, animal and human feces, and soil and plant samples (lactobacilli), or from environmental samples (*Bacillus*), and all strains belonged to the Culture Collections of Evonik Operations GmbH (*Bacillus*) and Universities of Bari and Bolzano (LABs). The cultivation of LABs was carried out in MRS medium at 30 °C for 20–24 h under anaerobic conditions. *Bacillus* strains underwent cultivation in LBG medium at pH 7.0 for 24 h at 37 °C under aerobic conditions.

### 2.2. Survival to Gastrointestinal Trait

Simulated gastric and intestinal fluids were obtained as described by Fernández et al. [27]. The harvesting of cells grown until the stationary phase was achieved by centrifugation at 8000× *g* for 10 min. Bacterial cells were washed with physiological solution (0.9% NaCl) and suspended in 50 mL of simulated gastric juice (9–10 log CFU/mL) containing NaCl (125 mM), KCl (7 mM), NaHCO_3_ (45 mM), and pepsin (3 g/L) (Sigma-Aldrich Co., St. Louis, MO, USA) [28]. Thus, the pH was corrected to 2.0, 3.0, and 8.0, as required, before incubating the suspension at 37 °C in anaerobic and stirring conditions. A pH value of 8.0 allowed us to investigate the effect of the components of the simulated gastric fluid on cells’ viability. The collection of aliquots from this suspension was carried out at 0, 90, and 180 min of incubation. At each sampled time, the cell density was estimated. Additionally, the influence of gastric digestion was also evaluated by reconstituted skim milk (RSM) (at a concentration of 11%; *w*/*v*) before the cells’ inoculation into simulated gastric juice at pH 2. The addiction of RSM determined a final pH equal to 3.0 ± 0.1. This condition simulates the influence of foods along the gastric transit [28]. At the end of gastric digestion (180 min), the cells were harvested and suspended in simulated intestinal juice containing pancreatin and Oxgall bile salt (Sigma-Aldrich Co.) (0.1% and 0.15% *w*/*v*, respectively). The pH was adjusted at 8.0. The incubation of the suspension was carried out at 37 °C under agitation and sampling was carried out at 0, 90, and 180 min [29]. Of the 504 assessed strains, 131 showed decreases of less than 2 log with respect to their initial cell density (9–10 log CFU/mL), which allowed us to define them as resistant to simulated gastrointestinal conditions. Resistant strains underwent further screening (Appendix A).

### 2.3. Peptidase Activities

Cultures collected from the late exponential phase of growth (ca. 9.0 log CFU/mL) underwent enzyme activity assays. Cell pellets (0.3 g (dry weight)) were washed (0.9% of NaCl; *w*/*v*), resuspended in 50 mM Tris–HCl (pH 7.0), and incubated for 30 min at 30 °C. In order to remove enzymes loosely associated to the cell wall, the suspension was centrifuged at 13,000× *g* for 10 min. In order to obtain the cytoplasmic extract, a lysozyme buffer containing 50 mM Tris–HCl (pH 7.5) and 24% of sucrose were added to the bacterial suspension before carrying out an incubation at 37 °C for 60 min while stirring (ca. 160 rpm). Spheroplasts were resuspended in isotonic buffer before the sonication process (40 s at 16 A/s) performed using the Sony Prep model 150 (Sanyo, UK). After a 10-fold-concentration by freeze-drying, cytoplasmic extracts were resuspended in 5 mM Tris-HCl (pH 7.0) and dialyzed (24 h; 4 °C). General aminopeptidase type N (PepN), proline iminopeptidase (PepI), X-prolyl dipeptidyl aminopeptidase (PepX), endopeptidase (PepO), and prolyl endopeptidase (PepP) activities from the cytoplasmic extracts of LABs and *Bacillus* were measured using Leu-p-nitroanilides (p-NA), Pro-p-NA, Gly-Pro-p-NA, Z-Gly-Gly-Leu-p-NA, and Z-Gly-Pro-p-NA (Sigma Chemical Co.; Saint Louis, United States), respectively. The mixture used to evaluate peptidase activity was obtained using 900 μL of 2.0 mM substrate in 0.05 M of potassium phosphate buffer (pH 7.0) and 100 μL of each cytoplasmic extract. After an incubation at 37 °C for 180 min, the absorbance was measured (wavelength 410 nm). A p-NA standard was used to compare the obtained data. One unit of activity was defined as the concentration of enzyme required to release 1 μmol/min of p-NA.

### 2.4. Hydrolysis of Immunogenic Epitopes

Cytoplasmic extracts of *Bacillus*, lactobacilli, and *Pediococcus* strains, with very high peptidase activities (at least for one peptidase), were pooled with the aim of combining intense and complementary enzyme activities. Each pooled cytoplasmic extract was used to assay its capacity to degrade immunogenic gluten epitopes in vitro. Chemically synthesized immunogenic epitopes were used at an initial concentration of 1 mM. The used immunogenic epitopes were 57–68 of α9-gliadin (Q-L-Q-P-F-P-Q-P-Q-L-P-Y), 62–75 of A-gliadin (P-Q-P-Q-L-P-Y-P-Q-P-Q-S-F-P), 134–153 of γ-gliadin (Q-Q-L-P-Q-P-Q-Q-P-Q-Q-S-F-P-Q-Q-Q-R-P-F), and 57–89 of α2-gliadin (33-mer; L-Q-L-Q-P-F-P-Q-P-Q-L-P-Y-P-Q-P-Q-L-P-Y-P-Q-P-Q-L-P-Y-P-Q-P-Q-P-F). 

Hydrolysis was monitored by RP-HPLC and the obtained peaks were singularly analyzed by nano-ESI–tandem mass spectrometry [30].

### 2.5. Gluten Degradation under Simulated Gastrointestinal Conditions

The gluten degradation under simulated gastrointestinal digestion was assessed. Based on the peptidase activities of pooled cytoplasmic extracts, microbial consortia were prepared using whole living and lysed cells of selected *Bacillus* and LABs strains. Five grams of wheat bread (chewed for 30 s) was placed into a beaker containing 10 mL of NaK-phosphate (0.05 M, pH 6.9) and related dough were suspended in simulated gastric juice containing NaCl (125 mM), KCl (7 mM), NaHCO_3_ (45 mM), and pepsin (3 g/L) (Sigma-Aldrich Co.). The suspension was added to the pooled selected strains as live (9.0 log CFU/g) and lysed bacteria (corresponding to 9.0 log cells/mL). A control, without the addition of bacterial cells, was also subjected to simulated digestion. The mixture was incubated at 37 °C, while stirring, to simulate peristalsis. After incubation (180 min), the suspension was added to 0.1% (*w*/*v*) pancreatin and 0.15% (*w*/*v*) Oxgall bile salt (Sigma-Aldrich Co.) at pH 8.0. Apart from pancreatin and bile salt, the fluid contained enzymatic preparations E1, E2 (BIO-CAT Inc., Troy, VA, USA; each 0.2 g/kg), Veron HPP (10 g/100 kg of protein), and Veron PS (AB Enzymes) (25 g/100 kg of protein) enzymes. Enzymatic preparations (E1, E2, Veron PS, and Veron HPP) were not added to the control dough. Intestinal digestion was carried out for 48 h at 37 °C under stirring conditions (ca. 200 rpm). After digestion, samples were placed on ice and the concentration of gluten was determined using an R5 antibody-based sandwich and competitive ELISA (R5-ELISA). R5-ELISA analysis was performed by using the RIDASCREEN^®^ Gliadin competitive detection kit (R-Biopharm AG; Pfungstadt, Germany). Moreover, an ELISA Systems Gluten Residue Detection Kit (Windsor, Australia) was used for the quantification of residual gluten. The presence of epitopes in digested samples was monitored via HPLC analysis after 6, 16, 24, 36, and 48 h of incubation. HPLC coupled with nano-ESI–tandem mass spectrometry was also performed to estimate the hydrolysis of gluten and the absence of toxic epitopes [30].

### 2.6. Immunogenicity Estimation of Gluten Digests Using Duodenal Explants from Celiac Disease Patients

The immunogenicity of the digests was estimated ex vivo by testing the level of interleukin 2 (IL-2), interleukin 10 (IL-10), and interferon gamma (IFN-γ) in duodenal biopsies of ten CD subjects. The study was conducted according to the guidelines of the Declaration of Helsinki and after the approval by the Ethics Committee of Ente Ospedaliero Specializzato in Gastroenterolologia IRCCS Saverio de Bellis (protocol code 1172012, 2012). Informed consent was obtained from all subjects involved in the study. All the ten celiac patients expressed the HLA-DQ2 phenotype and CD was diagnosed according to European Society for Pediatric Gastroenterology, Hepatology, and Nutrition criteria [31]. After excision, all biopsies were placed in ice-chilled medium (RPMI 1640; Gibco Invitrogen, Paisley, UK). Biopsies were incubated for 4 h as reported by Browning and Trier [32]. In detail, the biopsies were deposited villous side up on a stainless-steel mesh and disposed over the central well of an organ tissue culture dish (Falcon, Glendale, Arizona, USA). Each well was added of RPMI containing 15% newborn calf serum (Gibco Invitrogen) and 1% of antibiotics (penicillin and streptomycin) (Gibco Invitrogen, UK). Dishes were incubated at 37 °C under anaerobic conditions.

### 2.7. Statistical Analyses

All experiments were performed as biological triplicate with the only exception for the immunogenicity test, which was performed on a biological duplicate. The statistical software *Statistica* for Windows (Statistica 7.0 per Windows) was used to analyze collected data by one-way ANOVA and the average of paired comparison of treatment by Tukey’s procedure. In both statistical analyses, *p*-values < 0.05 indicating a statistically significant difference. The Statistica 7.0 software was also used to analyze the enzymatic activity data by principal component analysis (PCA).

## 3. Results

### 3.1. Selection of Bacterial Strains Resistant to Gastrointestinal Conditions

The initial screening of the 504 bacterial strains relied on their resistance to in vitro gastrointestinal conditions. The gastrointestinal resistance was evaluated in triplicate for each strain. Compared to *Weissella* and other LABs (*Leuconostoc* and *Pediococcus*), strains of *Bacillus* and lactobacilli showed higher (*p* < 0.05) median values of viable cell count (vcc) after in vitro gastrointestinal treatment at pH 2 (Figure 1).

At pH 2, *Weissella* had the value of the 75th vcc percentile, which was similar (*p* > 0.05) to the values for *Bacillus* and lactobacilli. The value of the 25th vcc percentile was lower, which indicated that more strains were less resistant to gastrointestinal conditions. *Bacillus* and lactobacilli strains had the highest value of the 25th vcc percentile. Within lactobacilli, *Lp. plantarum* and *Lc. paracasei* were the most resistant species. As indicated by median vcc values, *Lp. plantarum* and *Lc. paracasei* strains lost only two cycles (log10) during gastric treatment at pH 2 (Appendix A). *Lc. paracasei* had both the highest median value and the highest value of the 25th vcc percentile. *Lp. plantarum* strains behaved similarly (Appendix A). Meanwhile, within *Bacillus* (*B.*) species, for *B. licheniformis* strains, which showed the highest value of the 75th vcc percentile (close to 9 CFU/mL) after gastric digestion, this result was not confirmed after intestinal digestion had also taken place (Appendix A). In fact, when looking at the vcc after simulated intestinal digestion, we found that the best survival strains belong to *B. pumilus* and *B. velezensis*. These groups showed the highest median and 25th vcc percentile in all four tested intestinal conditions (pH2, pH3, pH2SM, and pH8) (Appendix A). 

Overall, compared to pH 2, all strains showed an increased survival at pH 2 plus RSM and at pH 3. The treatment at pH 8 was used as a positive control. Hence, 64 out of 190 *Bacillus* strains (33.7%), 63 out of 285 lactobacilli strains (22.1%), 1 out of 5 *Pediococcus* strains (20%), and 3 out of 18 *Weissella* strains (16.7%) showed the highest resistance to gastrointestinal conditions and were selected for the further screening based on their peptidase activities.

### 3.2. Selection of Bacterial Strains Based on Peptidase Activities

The 131 *Bacillus* and LABs strains were further assessed for PepN, PepI, PepX, PepO, and PepP activities using Leu-p-NA, Pro-p-NA, Gly-Pro-p-NA, Z-Gly-Gly-Leu-p-NA, and Z-Gly-Pro-4-nitroanilide synthetic substrates, respectively. All assays used the cytoplasmic extract from each selected strain [30]. Some strains could be clearly distinguished based on principal component analysis (PCA) of peptidase activity data (Figure 2). According to the loading plot (Figure 2B), strains with the lowest peptidase activities are located in the negative fourth quadrant (negative PC1 and PC2) (Figure 2A). By contrast, strains showing the highest activities toward Leu-p-NA, Pro-p-NA, Gly-Pro-p-NA, and Z-Gly-Pro-4-nitroanilide substrates are distributed in the first (positive PC1 and PC2) and second (positive PC1, negative PC2) quadrants. Strains with the highest hydrolytic activity toward Z-Gly-Gly-Leu-p-NA are mainly distributed in the third quadrant (negative PC1 and positive PC2).

The 24 strains showing very high peptidase activities (at least toward one synthetic substrate) (Appendix A) were deposited in the Leibniz Institute DSMZ—German Collection of Microorganisms and Cell Cultures GmbH (Deutsche Sammlung von Mikroorganismen und Zellkulturen GmbH) with a new code (Appendix A; Figure 2 and Appendix A). Detailing each peptidase activity, PepN ranged from 0.0 (*B. subtilis* U002-C04, *B. megaterium* U541-C05, *B. velezensis* U021-C01, and *B. pumilus* DSM33301) to 31.40 ± 0.09 U (*Lp. plantarum* DSM33362), with a median value of 3.08 U. The other most active strains were *B. licheniformis* DSM33354, *B. megaterium* DSM33356, *B. pumilus* DSM33297, *B. subtilis* DSM33298, *Lv. brevis* DSM33377, *Lp. plantarum* (DSM33367 and DSM33370, and DSM33363), *Lc. paracasei* (DSM33373, DSM33376, and DSM33375), *Ls. reuteri* DSM33374, and *Ped. pentosaceus* DSM33371 (Figure 2 and Appendix A). The median value for PepI activity was of 1.66 U. The most active strains (PepI activity > 18 U) were of *Lc. paracasei* (DSM33375 and DSM33373). PepX activity ranged from 0.0 to ca. 24 U. Compared to the median value of 1.81 U, *Lc. paracasei* DSM33373, *Lp. plantarum* (DSM33370, DSM33369, and DSM33363), *Ls. reuteri* DSM33374, *Fr. sanfranciscensis* DSM33379, and *Ped. pentosaceus* DSM33371 showed the highest activities. The median value for PepO activity was set at 0.54 U. Strains showing activities higher than 5 U were *B. subtilis* DSM33353 and *B. pumilus* DSM33355 and DSM33301. PepP activity varied from 0.0 to 6.23 U (*Lp. plantarum* DSM33368). The median value was 0.22 U. *B. megaterium* DSM33300, *Lv. brevis* DSM33377, *Lc. paracasei* DSM33373, *Lp. plantarum* DSM33364, DSM33366, and DSM33367, *Ls. reuteri* DSM33374, *Fr. sanfranciscensis* DSM33378, and *Ped. pentosaceus* DSM33371 had activities higher than 3 U. The sequential screenings based on resistance to simulated gastrointestinal conditions and peptidase activities made the selection of 24 strains possible, which comprised the various microbial consortia used for hydrolyzing gluten and gluten immunogenic epitopes. 

### 3.3. Hydrolysis of Gluten Immunogenic Epitopes 

The ability of the selected strains to hydrolyze gluten immunogenic epitopes was tested using cytoplasmic extracts. Combining complementary hydrolyzing activities, six pooled cytoplasmic extracts (CE1–CE6) were prepared (Figure 3 and Appendix A) to assay their capability to hydrolyze gluten immunogenic epitopes in vitro. The hydrolysis of immunogenic epitopes, specifically 57–68 of α9-gliadin, 62–75 of A-gliadin, 134–153 of γ-gliadin, and 57–89 (33-mer) of α2-gliadin, was determined by RP-HPLC and nano-ESI–MS/MS. The analysis was carried out in biological triplicate. All six cytoplasmic extracts had the capability to decrease the concentration of each gluten immunogenic epitope. Nevertheless, only CE3 and CE4 fully hydrolyzed all the epitopes. However, CE1, CE2, CE5, and CE6 strongly reduced the concentration of all gluten epitopes. Compared to the initial concentration (1 mM), the residual amount of each epitope decreased by at least 70%. Some RP-HPLC chromatograms showing total or partial hydrolysis of the gluten immunogenic epitopes were reported in Appendix A.

### 3.4. Gluten Degradation

The strains comprising the CE3 and CE4 enzyme preparations were used to hydrolyze gluten under simulated gastrointestinal digestion of wheat bread. The following combinations were tested with three independent analyses:
C3 consortium: *Lp. plantarum* DSM33370, DSM33363, and DSM33364; *Lc. paracasei* DSM33373; *Lv. brevis* DSM33377; *B. pumilus* DSM33297 and DSM33355; *B. licheniformis* DSM33354; *B. megaterium* DSM33300; and *B. subtilis* DSM33353.C4 consortium: *Lp. plantarum* DSM33362, DSM33367, and DSM33368; *Lc. paracasei* DSM33375; *Fr. sanfranciscensis* DSM33379; *B. pumilus* DSM33301; *B. megaterium* DSM33300 and DSM33356; and *B. subtilis* DSM33298 and DSM33353.


For this assay, all strains were used as whole living cells, alone or together with the corresponding lysed cells. Whole cells were used to evaluate the probiotic activity (gluten degradation) at gut level [33]. Gluten hydrolysis was estimated using both the sandwich and competitive ELISA assays. Sandwich ELISA was used to quantify residual gluten, while ELISA RIDASCREEN^®^ Gliadin competitive was used to quantify peptide fragments of prolamins (AOAC approved Official Method of Analysis, Final Action OMA 2015.05). Compared to the control, the addition of selected strains significantly decreased the amount of gluten and the peptide fragments of prolamins (data not shown). The addition of lysed cells enhanced gluten degradation. However, the gluten and peptide fragments of prolamins were still detectable at the end of digestion.

To accelerate gluten hydrolysis, 16 different microbial consortia were constructed by combining the complementary peptidase activities of the 24 selected strains and bacterial and fungal commercial enzymes (E1, E2, Veron HPP, and Veron PS proteases) [34] (Table 1). 

Consortia sizes were lower than the C3 and C4 consortia in order to determine the strains with critical functionality and to allow for a more streamlined prototype development for subsequent clinical testing. As estimated by sandwich ELISA, the concentration of gluten after 6 h of digestion ranged from ca. 1100 (control) to <20 ppm (Consortia 8 and 16) (Table 1). After 6 h, microbial Consortia 4 and 9 showed a markedly decreased gluten content (190 ± 0.05 and 60 ± 0.02 ppm, respectively). Extending the incubation to 16 and 24 h, gluten was not detectable when digested by MC4, MC8, MC9, and MC16. After incubation, residual gluten was only detectable in the presence of four microbial consortia (MC11, MC12, MC14, and MC15). 

As estimated by competitive ELISA, the concentration of peptide fragments derived from prolamins after 6 h of incubation ranged from ca. 810 (control) to 280 ppm (MC16) (Table 1). Compared to other consortia, only MC4 and MC16 showed the full hydrolysis of peptide fragments of prolamins after 36 h of digestion. Further extending the time of incubation (48 h), the complete removal of peptide fragments of prolamins was also observed for Consortia 5, 6, and 8. The ability of E1, E2, Veron HPP, and Veron PS proteases to hydrolyze gluten without bacterial cells was also estimated (Appendix A). The complete hydrolysis of gluten and related peptides was obtained only by using selected microbial consortia and proteases.

### 3.5. Gluten Digest Immunogenicity toward Duodenal Explants

Cultures of duodenal explants from celiac disease patients serve as an effective preclinical tool for probing the possible immunogenicity of gluten digests. Such a test therefore gives valuable insights into the safety and efficacy of the gluten metabolism interventions not only as addressed to these patients but also for other gluten-related disorders. Wheat bread digested without the addition of bacterial cells and microbial enzymes (positive control), using microbial consortia with medium (MC7), or the highest (MC4 and MC16) hydrolytic activity against gluten was tested on duodenal biopsy specimens from patients with CD. As expected, the duodenal biopsy specimens incubated with the positive control favored the significantly (*p* < 0.05) higher expression of interleukin 2 (IL-2), interleukin 10 (IL-10), and interferon gamma (IFN-γ) mRNA with respect to the negative control (RPMI 1640 + gastric and intestinal juice) (Figure 4). Contrarily, specimens in which MC4 and MC16 digested gluten did not show any significant differences in the level of expression of IL-2, IL-10, and IFN-γ compared to the negative control. Consortium 7 favored the synthesis of IL-2, IL-10, and IFN-γ even if lower than the positive control. However, the levels of IL-2, IL-10, and IFN-γ expressed in CD biopsies after adding MC7 were higher than how MC4 and MC16 determined it. 

## 4. Discussion

Using a large number of strains (504) previously isolated from the human intestine or food matrices and environmental samples, this study preliminarily selected 131 strains belonging to species of *Bacillus* (64), lactobacilli (63), *Pediococcus* (1), and *Weissella* (3) which showed marked resistance to in vitro gastrointestinal conditions. All of the selected strains showed a cell survival higher than 7 log CFU/mL, thus being suitable candidates to hydrolyze gluten immunogenic epitopes under gastrointestinal conditions. First, the selected strains were assessed for complementary peptidase activities (PepN, PepI, PepX, PepO, and PepP), which are potentially responsible for the hydrolysis of 33-mer and other gluten immunogenic epitopes (57–68 of α9-gliadin, 62–75 of A-gliadin, and 134–153 of γ-gliadin). A previous report [35] stated that combined peptidase activities (general aminopeptidase type N, endopeptidase, and prolyl endopeptidyl peptidase) improved the hydrolysis of CD epitopes. The combination of general aminopeptidase type N and X-prolyl dipeptidyl aminopeptidase liberated non-immunogenic dipeptides from 33-mer [30]. Other authors reported that the hydrolysis of synthetic gliadin peptides by *Lactobacillus paracasei* (reassigned as *Lacticaseibacillus paracasei*) interfered with their entrance into the epithelial compartment [36]. A mixture of *L. casei* (reassigned as *Lacticaseibacillus casei*), *L. plantarum* (reassigned as *Lactiplantibacillus plantarum*), *Bifidobacterium animalis* subsp. *lactis*, and *Bifidobacterium breve* decreased the in vitro toxicity of gliadins by hydrolyzing some immune-dominant peptides [37]. 

The multivariate statistical analysis we used (PCA), based on peptidase activity, clearly distinguished 24 strains belonging to *Lp. plantarum*, *Ls. reuteri, Lc. paracasei*, *Lv. brevis*, *Fr*. *sanfranciscensis*, *Ped*. *pentosaceus*, *B. pumilus*, *B. licheniformis*, *B. megaterium*, and *B. subtilis*, all of which exhibited the highest peptidase activities compared to the other tested strains. As estimated by RP-HPLC and nano-ESI–MS/MS, cytoplasmic enzymes of the selected strains had the capability to hydrolyze several gluten immunogenic epitopes, specifically 57–68 of α9-gliadin, 62–75 of A-gliadin, 134–153 of γ-gliadin, and 57–89 (33-mer) of α2-gliadin. Numerous in vitro studies showed that human commensal bacteria might affect the digestion of gliadins [26,38,39]. A microbial consortium comprising 27 bacterial strains, belonging to *L. salivarius* (reassigned as *Ligilactobacillus salivarius*), *L. rhamnosus* (reassigned as *Lacticaseibacillus rhamnosus*), *L. reuteri* (reassigned as *Limosilactobacillus reuteri*), *L. casei* (reassigned as *Lacticaseibacillus casei*), *L. oris* (reassigned as *Limosilactobacillus oris*), *L. gasseri*, *L. fermentum* (reassigned as *Limosilactobacillus fermentum*), *L. crispatus*, *L. brevis* (reassigned as *Levilactobacillus brevis*), *B. subtilis*, *B. amyloliquefaciens*, *B. pumilus*, and *B. licheniformis* isolated from the human small intestine, hydrolyzed the 33-mer epitope in 24 h [14]. Similar data were obtained for other strains of human small intestinal origin, including of the species *B. subtilis*, *B. pumilus*, and *B. licheniformis* [40]. Importantly, (small intestinal) microbiota composition and activity are a critical determinant of gluten metabolism, as exemplified by Caminero and coworkers in a comparison of gluten digests produced by microbiota derived from CD patients and from healthy controls [41]. Interventions that interfere with gluten metabolism need to be tested under gastrointestinal conditions, including a complex microbiota, and ideally need to assess the possible immunogenicity of the resulting digests. During digestive processes, gluten undergoes digestion by pepsin and other human as well as microbial enzymes, which leads to free epitopes that are responsible for gluten-related disorders in susceptible individuals. Previously, the proteolytic cleavage of cereal (wheat, barley, rye, triticale, and oat) gliadins was demonstrable using microbial proteases from *B. stearothermophilus*, *B. thermoproteolyticus*, *B. licheniformis*, and *Streptomyces griseus* [24]. Along the same lines, such glutenases and other endopeptidases that destroy gluten epitopes have been proposed as enzymatic supplements, which in combination with GFD, might eliminate the toxicity of accidental gluten exposure. The novel pepsin-resistant endoprotease 40 (E40 glutenase) was recently discovered [18] as a secreted protease from the acidophilic actinomycete *Actinoallomurus* A8 (International patent application WO 2013/083338). Plant- and microorganism-based prolyl endopeptidases (PEPs) have been proposed to supplement the inadequate repertoire of human digestive enzymes [19]. Fungal proteases are also promising sources. The most effective proteolytic activity was detectable using the acid proteinase from *A. niger,* which led to the full degradation of wheat gliadins and low molecular weight peptides [14,42]. Even if wheat gliadins are susceptible to enzymatic degradation, further immunochemical or mass spectrometry analyses are desirable in order to confirm whether the products of proteolysis have been lost or at least experienced a partial decrease in their immunogenic activities. Given the diversity of gluten-inherent peptide sequences with immunogenic potential, a combination of peptide hydrolases from different microbes is required to ensure complete degradation of all potential peptides. In addition, some gluten-hydrolyzing enzymes do not optimally function at the acid pH of the stomach and are susceptible to pepsin digestion [43]. A microbial consortium comprising ten selected lactic acid bacteria strains and fungal proteases fully hydrolyzed wheat gluten during processing and allowed for the manufacture of gluten-free leavened baked goods from wheat flour which were tolerated by celiac patients [44]. In light of this scientific evidence, the use of microbial consortia selected for their ability to hydrolyze gluten epitopes under human gastrointestinal conditions together with gluten-hydrolyzing enzymes could be a new strategy to improve, or at least maintain, the health of patients with gluten-related disorders. For the first time, this study combined free bacterial and fungal proteolytic enzymes (*A. oryzae* E1, *A. niger* E2, and *B. subtilis* Veron HPP and Veron PS proteases) with microbial consortia, combining lactobacilli and *Bacillus* species, that showed high gastric and bile resistance and could hydrolyze gluten during gastrointestinal digestion. As estimated by ELISA tests and RP-HPLC coupled with nano-ESI–MS/MS, 2 microbial consortia (Consortium 4: *Lp. plantarum* DSM33363 and DSM33364, *Lc. paracasei* DSM33373, *B. subtilis* DSM33298, and *B. pumilus* DSM33301; and Consortium 16: *Lp. plantarum* DSM33363 and DSM33364, *Lc. paracasei* DSM33373, *Ls. reuteri* DSM33374, *B. megaterium* DSM33300, and *B. pumilus* DSM33297 and DSM33355) containing commercial enzymes were able to hydrolyze gluten and its related immunogenic epitopes under gastrointestinal conditions. Despite the results obtained by ELISA tests and nano-ESI–MS/MS, the detection of all immunogenic epitopes may be incomplete. Since the presence of unknown immunogenic peptides cannot be excluded, cytokine expression in duodenal biopsy specimens from CD patients [43] was studied. Gluten peptides induced cytokine expression (e.g., IL-2, IL-10, IFN-γ), which was correlated with histological changes in the small intestinal mucosa in CD patients [45]. Recently, it was shown that the administration of gluten peptides in CD patients drives a rapid increase in circulating cytokines, including IL-2 and IL-10, which was closely associated with the onset and severity of acute digestive symptoms [46]. As shown using duodenal biopsy specimens, wheat bread digested by selected microbial consortia (4 or 16) loses its ability to induce IL-2, IL-10, and IFN-γ expression. IFN-γ-blocking antibodies prevented damage to the intestinal mucosa when exposed to inflammatory cytokines released by gliadin-specific T-cell lines [47]. It was shown that gliadin-induced-IFN-γ secretion increases gliadin influx through the intestinal barrier, leading to mucosal injury and the atrophy of villi [47]. In the present paper, we tested the pro-inflammatory activity of gluten (and related) peptides on celiac patients’ biopsies, confirming their role as a trigger of gluten mediated mucosal damage. The chance of testing the pro-inflammatory effect in the settings of a gluten-related condition is limited in animal models, and although different animal models have recently been created to replicate various aspects of the complex CD pathogenesis. The diverse associations between the innate and adaptive immune responses to gluten and gluten-dependent autoimmunity in CD are only now being discussed in mouse strains. We have learned that innate and adaptive gluten sensitivity and autoimmunity are simultaneously operating in CD activation due to environmental factors such as breastfeeding duration, gluten introduction and quantity, and viral diseases in the weaning period. Since these risk factors have not yet been integrated into the existing celiac disease animal models, it may be beneficial. In previous work, C57BL/6 mice were immunized with gliadin to activate gluten specific CD4+ T cells. Subsequently to administering a gluten-enriched and a gluten-free diet, gluten-sensitized mice developed severe duodenal epithelial lesions. The authors observed that the mucosal damage was exacerbated by a gluten-containing diet that induced significant weight loss, duodenitis, and enhanced levels of IFN-γ and interleukin-17 duodenal transcripts [48,49]. Although these results are preliminary, it is fascinating to speculate that mouse models may serve as a tool in the evaluation of treatment opportunities for CD, which ultimately need to be tested in humans. 

## 5. Conclusions

This study provides evidence that the selected microbial consortia and microbial proteolytic enzymes have the potential to improve the digestion of gluten and to hydrolyze immunogenic peptides during gastrointestinal digestion. Further studies are warranted to provide deeper insights into the effects of these microbial consortia and proteolytic enzymes on in vivo gluten digestion. Another important result of this work is the identification of a large number of acid-resistant lactic acid bacteria and *Bacillus* strains that could be applied in food biotechnologies to partially or totally hydrolyze gluten in cereal-based products.

## Figures and Tables

**Figure 1 nutrients-13-00992-f001:**
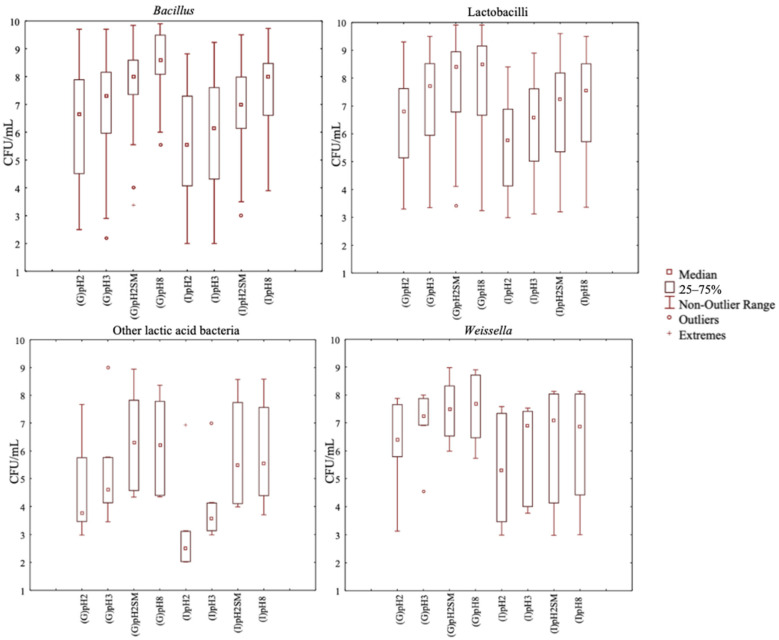
Viable cell count (CFU/mL) of the bacterial strains tested for their resistance to the gastro (G) and then intestinal (I) transit in vitro at different conditions: pH 2 (pH2), pH 3 (pH3), pH 2 plus skim milk (pH2SM), and pH 8 (pH8). All strains belong to *Bacillus*, lactobacilli, *Weissella*, *Leuconostoc*, and *Pediococcus* species. The group named “other lactic acid bacteria” includes *Leuconostoc* and *Pediococcus* strains.

**Figure 2 nutrients-13-00992-f002:**
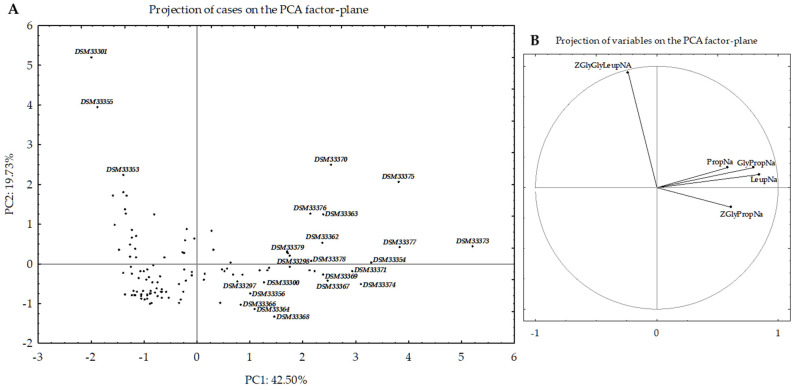
Score (**A**) and loading (**B**) plots of the first and second principal components (PC) after principal component analysis (PCA) based on the general aminopeptidase type N (PepN), proline iminopeptidase (PepI), X-prolyl dipeptidyl aminopeptidase (PepX), endopeptidase (PepO), and prolyl endopeptidase (PepP) activities of the cytoplasmic extracts of the 131 *Bacillus*, lactobacilli, and *Weissella* strains. PepN, PepI, PepX, and PepP were measured using Leu-p-nitroanilides (p-NA), Pro-p-NA, Gly-Pro-p-NA, Z-Gly-Gly-Leu-p-NA, and Z-Gly-Pro-4-nitroanilide substrates, respectively. All strains are reported in the plot by circles and the strain code was reported only for the selected strains.

**Figure 3 nutrients-13-00992-f003:**
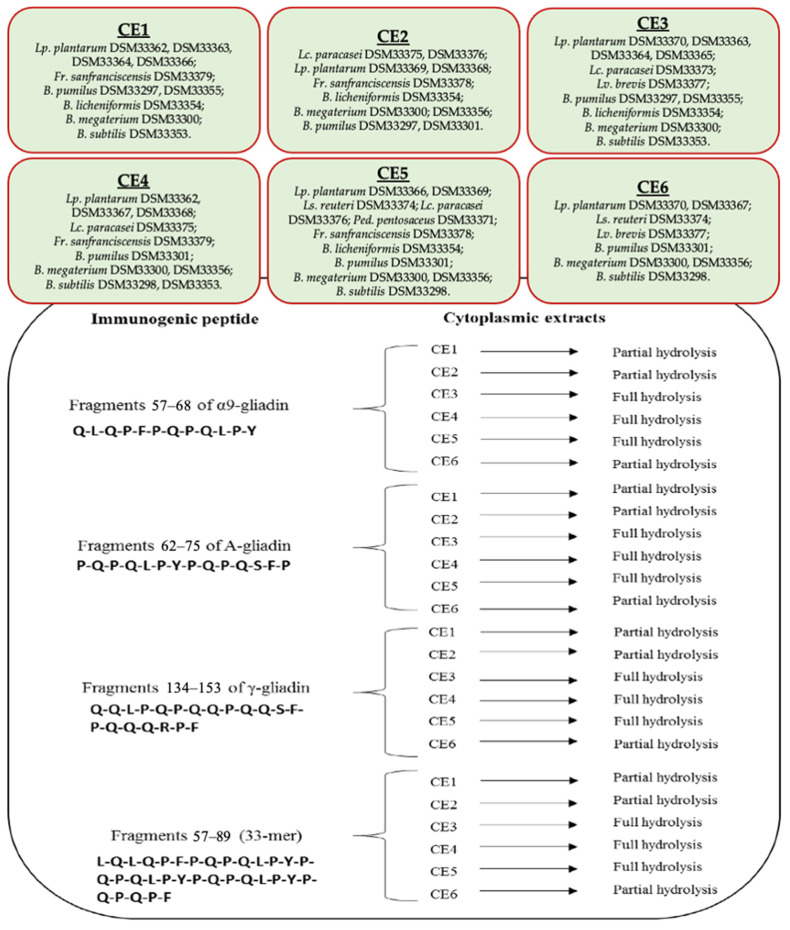
The tested bacterial cytoplasmic extracts (CE1–CE6) and the related peptidase activities against immunogenic epitopes.

**Figure 4 nutrients-13-00992-f004:**
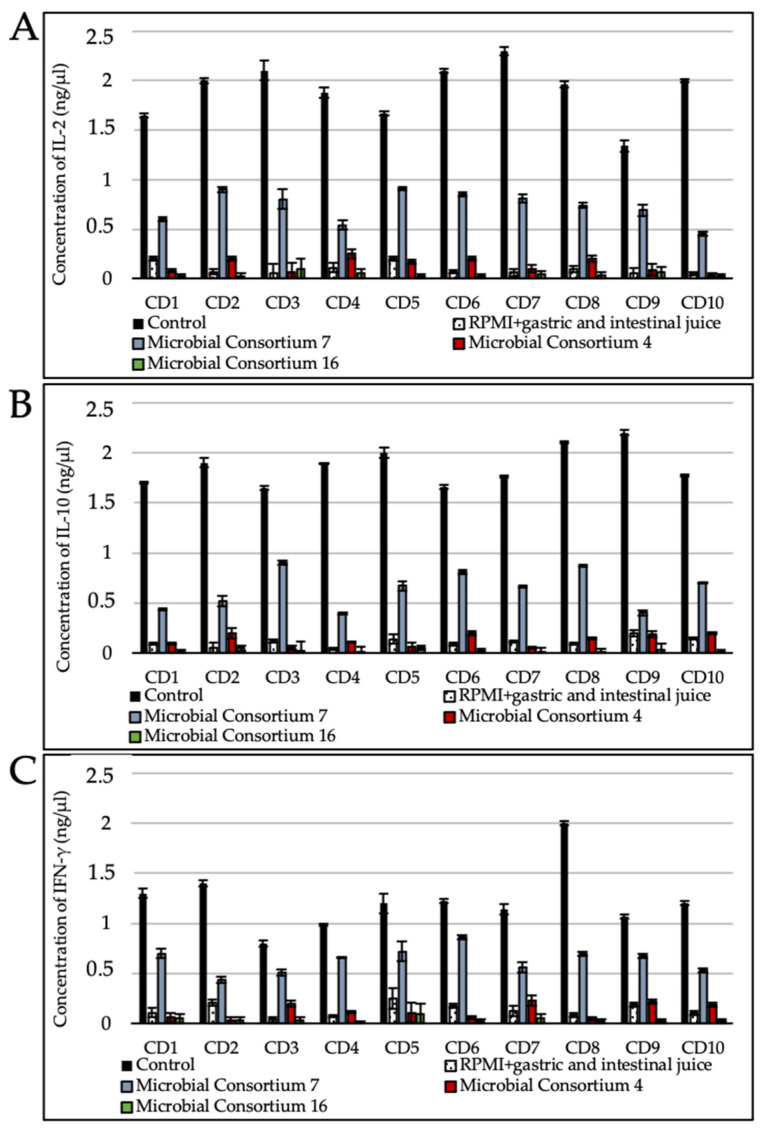
Concentration (ng/µL) of interleukin 2 (IL-2) (**A**), interleukin 10 (IL-10) (**B**), and interferon gamma (IFN-γ) (**C**) in duodenal biopsy specimens from patients with CD. Analyses were carried out in two independent experiments. Control, wheat bread digested without the addition of bacterial cells and microbial enzymes; RPMI+gastric and intestinal juice, negative control; Consortium 4, wheat bread digested with the addition of live and lysed cells of *Lp. plantarum* DSM33363 and DSM33364, *Lc. paracasei* DSM33373, *B. subtilis* DSM33298, and *B. pumilus* DSM33301 and E1, E2, Veron PS, and Veron HPP commercial enzymes); Consortium 7, wheat bread digested with the addition of live and lysed cells of *Lp. plantarum* DSM33362, DSM33366, and DSM33370, *Ls. reuteri* DSM33374, *B. megaterium* DSM33356, and *B. subtilis* DSM33353 and E1, E2, Veron PS, and Veron HPP commercial enzymes; and Consortium 16, wheat bread digested with the addition of live and lysed cells of *Lp. plantarum* DSM33363 and DSM33364, *Lc. paracasei* DSM33373, *Ls. reuteri* DSM33374, *B. megaterium* DSM33300, *B. pumilus* DSM33297 and DSM33355 and E1, E2, Veron PS, and Veron HPP commercial enzymes. CD1 to CD10, duodenal biopsy specimens from celiac patients. The evaluation was conducted on biological duplicates and the mean values (± standard deviation) were reported.

**Table 1 nutrients-13-00992-t001:** Concentration (ppm) of residual gluten and peptide fragments of prolamins after 6, 16, 24, 36, and 48 h of simulated gastrointestinal digestion, as estimated by specific ELISA tests. Control: dough digested without bacterial cells and commercial enzymes; MC1–MC16: microbial consortia constructed by using live and lysed cells of selected lactobacilli (*Lactiplantibacillus*, *Lp*.; *Lacticaseibacillus*, *Lc*.; *Levilactobacillus*, *Lv*.; *Limosilactobacillus*, *Ls*., *Fructilactobacillus*, *Fr*.; *Pediococcus*, *Ped*.) and *Bacillus* (B.) strains and E1, E2, Veron PS, and Veron HPP commercial enzymes.

Strains		Sandwich ELISA Assay (Residual Gluten)					Competitive ELISA Assay (Peptide Fragments)				
		6 h	16 h	24 h	36 h	48 h	6 h	16 h	24 h	36 h	48 h
Control		1100 ^a^ ± 0.06	620 ^a^ ± 0.09	367 ^a^ ± 0.05	256 ^a^ ± 0.04	75 ^a^ ± 0.06	810 ^a^ ± 0.03	400 ^a^ ± 0.02	397 ^a^ ± 0.08	381 ^a^ ± 0.07	375 ^a^ ± 0.05
MC1	*Lp. plantarum* DSM33370, DSM33363, DSM33364; *Lc. paracasei* DSM33373; *Lv. brevis* DSM33377; *B. pumilus* DSM33297, DSM33355, DSM33301	406 ^b^ ± 0.04	135 ^b^ ± 0.06	19 ^e^ ± 0.01	0 ^e^	0 ^e^	310 ^f^ ± 0.05	250 ^d^ ± 0.03	200 ^e^ ± 0.04	170 ^e^ ± 0.02	65 ^g^ ± 0.01
MC2	*Lp. plantarum* DSM33362, DSM33367, DSM33368; *Lc. paracasei* DSM33375; *B. subtilis* DSM33298; *B. licheniformis* DSM33354; *B. megaterium* DSM33300	346 ^c^ ± 0.07	121 ^b^ ± 0.03	15 ^e^ ± 0.01	0 ^e^	0 ^e^	332 ^f^ ± 0.05	226 ^ef^ ± 0.04	167 ^f^ ± 0.03	158 ^e^ ± 0.02	150 ^c^ ± 0.02
MC3	*Lp. plantarum* DSM33366, DSM33369; *Ls. reuteri* DSM33374; *Lc. paracasei* DSM33376; *Ped. pentosaceus* DSM33371; *B. megaterium* DSM33356; *B. subtilis* DSM33353	382 ^c^ ± 0.03	99 ^c^ ± 0.02	12 ^f^ ± 0.01	0 ^e^	0 ^e^	315 ^f^ ± 0.06	272 ^d^ ± 0.07	256 ^d^ ± 0.04	244 ^c^ ± 0.05	228 ^b^ ± 0.02
MC4	*Lp. plantarum* DSM33363, DSM33364; *Lc. paracasei* DSM33373; *B. subtilis* DSM33298; *B. pumilus* DSM33301	190 ^cd^ ± 0.05	0 ^g^	0 ^g^	0 ^e^	0 ^e^	399 ^e^ ± 0.08	233 ^e^ ± 0.07	112 ^g^ ± 0.05	0 ^j^	0 ^g^
MC5	*Lv. brevis* DSM33377; *Ped. pentosaceus* DSM33371; *Lp. plantarum* DSM33369; *B. pumilus* DSM33297; *B. megaterium* DSM33300	380 ^b^ ± 0.06	18 ^e^ ± 0.01	5 ^g^ ± 0.01	0 ^e^	0 ^e^	398 ^e^ ± 0.04	221 ^e^ ± 0.05	154 ^fg^ ± 0.03	46 ^i^ ± 0.02	0 ^g^
MC6	*Lc. paracasei* DSM33375; *Lp. plantarum* DSM33367, DSM33368; *B. pumilus* DSM33355; *B. licheniformis* DSM33354	350 ^c^ ± 0.06	15 ^e^ ± 0.02	2 ^g^ ± 0.01	0 ^e^	0 ^e^	404 ^e^ ± 0.06	245 ^de^ ± 0.05	100 ^g^ ± 0.08	79 ^h^ ± 0.04	0 ^g^
MC7	*Lp. plantarum* DSM33370, DSM33362, DSM33366; *Ls. reuteri* DSM33374; *B. megaterium* DSM33356; *B. subtilis* DSM33353	360 ^c^ ± 0.09	20 ^e^ ± 0.06	10 ^f^ ± 0.01	0 ^e^	0 ^e^	401 ^e^ ± 0.07	261 ^d^ ± 0.05	150 ^f^ ± 0.03	99 ^g^ ± 0.04	78 ^g^ ± 0.05
MC8	*Lp. plantarum* DSM33363, DSM33364; *Lc. paracasei* DSM33375; *Ls. reuteri* DSM33374; *B. megaterium* DSM33300; *B. pumilus* DSM33297	18 ^g^ ± 0.03	3 ^g^ ± 0.01	0 ^g^	0 ^e^	0 ^e^	323 ^f^ ± 0.08	228 ^e^ ± 0.06	218 ^e^ ± 0.05	157 ^e^ ± 0.06	0 ^g^
MC9	*Lc. paracasei* DSM33375; *Lp. plantarum* DSM33367; *Ls. reuteri* DSM33374; *B. megaterium* DSM33300; *B. pumilus* DSM33297; *B. licheniformis* DSM33354	60 ^f^ ± 0.04	12 ^f^ ± 0.01	0 ^g^	0 ^e^	0 ^e^	319 ^f^ ± 0.06	211 ^f^ ± 0.05	196 ^ef^ ± 0.03	195 ^de^ ± 0.07	152 ^c^ ± 0.02
MC10	*Lp. plantarum* DSM33363, DSM33364, DSM33370; *Lv. brevis* DSM33377; *B. pumilus* DSM33297; *B. megaterium* DSM33356	112 ^e^ ± 0.06	77 ^d^ ± 0.04	70 ^d^ ± 0.02	0 ^e^	0 ^e^	465 ^d^ ± 0.09	370 ^b^ ± 0.06	243 ^de^ ± 0.05	145 ^ef^ ± 0.04	97 ^f^ ± 0.03
MC11	*Lp. plantarum* DSM33368, DSM33362, DSM33367; *Lc. paracasei* DSM33375; *B. megaterium* DSM33300; *B. subtilis* DSM33353	221 ^d^ ± 0.05	89 ^c^ ± 0.07	69 ^d^ ± 0.06	50 ^d^ ± 0.04	43 ^d^ ± 0.03	512 ^c^ ± 0.06	367 ^b^ ± 0.08	340 ^b^ ± 0.09	300 ^b^ ± 0.06	123 ^de^ ± 0.05
MC12	*Lp. plantarum* DSM33366, DSM33369; *Ls. reuteri* DSM33374; *Lc. paracasei* DSM33376; *Ped. pentosaceus* DSM33371; *B. pumilus* DSM33297, DSM33355	145 ^e^ ± 0.06	110 ^c^ ± 0.05	89 ^c^ ± 0.03	75 ^c^ ± 0.02	63 ^b^ ± 0.03	601 ^b^ ± 0.09	312 ^c^ ± 0.06	289 ^c^ ± 0.07	288 ^b^ ± 0.05	143 ^cd^ ± 0.03
MC13	*Lv. brevis* DSM33377; *Ped. pentosaceus* DSM33371; *Fr. sanfranciscensis* DSM33379; *B. megaterium* DSM33300; *B. pumilus* DSM33297	163 ^de^ ± 0.06	122 ^b^ ± 0.04	82 ^c^ ± 0.02	45 ^d^ ± 0.03	0 ^e^	523 ^c^ ± 0.07	322 ^c^ ± 0.07	321 ^b^ ± 0.06	215 ^d^ ± 0.07	134 ^d^ ± 0.05
MC14	*Lp. plantarum* DSM33368; *Lc. paracasei* DSM33375; *Fr. sanfranciscensis* DSM33378; *B. megaterium* DSM33300; *B. pumilus* DSM33297; *B. licheniformis* DSM33354	234 ^d^ ± 0.08	135 ^b^ ± 0.07	120 ^b^ ± 0.07	108 ^b^ ± 0.05	56 ^c^ ± 0.03	587 ^b^ ± 0.09	333 ^c^ ± 0.09	256 ^d^ ± 0.08	211 ^d^ ± 0.08	167 ^c^ ± 0.07
MC15	*Lp. plantarum* DSM33362, DSM33366, DSM33370; *Ls. reuteri* DSM33374; *Fr. sanfranciscensis* DSM33378, DSM33379; *B. licheniformis* DSM33354; *B. subtilis* DSM33353	199 ^d^ ± 0.05	100 ^c^ ± 0.04	81 ^c^ ± 0.05	59 ^d^ ± 0.04	40 ^d^ ± 0.03	498 ^c^ ± 0.08	318 ^c^ ± 0.04	280 ^c^ ± 0.03	256 ^c^ ± 0.08	118 ^e^ ± 0.05
MC16	*Lp. plantarum* DSM33363, DSM33364; *Lc. paracasei* DSM33373; *Ls. reuteri* DSM33374; *B. megaterium* DSM33300; *B. pumilus* DSM33297, DSM33355	19 ^g^ ± 0.03	11 ^f^ ± 0.01	0 ^g^	0 ^e^	0 ^e^	280 ^g^ ± 0.06	200 ^f^ ± 0.05	50 ^h^ ± 0.03	10 ^j^ ± 0.01	0 ^g^

Data are the mean of three independent biological replicates. ^(a–j)^ Values with different superscript letters within the same row indicate significant differences (*p* < 0.05).

## Data Availability

No new data were created or analyzed in this study. Data sharing is not applicable to this article.

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
