# Peer review of "Selection of Gut-Resistant Bacteria and Construction of Microbial Consortia for Improving Gluten Digestion under Simulated Gastrointestinal Conditions"

_nutrients, 2021, doi:10.3390/nu13030992_

Round 1
Reviewer 1 Report
This paper systematical explored strains of bacteria relevant to gluten peptide hydrolysis. It was interesting, well written, and generally clear. However, I do have a few concerns:
- The paragraph from lines 109-123 was confusing and difficult to read. It wasn't until the results section that I figured out exactly how many strains were actually tested, especially as the numbers listed in this paragraph are inconsistent. For example, stating that there were 190 strains of Bacillus but then listing a lot of other non-Bacillus without numbers in the subsequent subcategories lost me for quite a while.
- How many replicates of digestion were performed per strain?
- How many replicates were performed for the hydrolysis experiments? Based on the error bars for on the figures, I am guessing at least some replicates were performed, but there is no mention of how many.
- Again, how many replicates for the gluten degradation under simulated gastrointestinal conditions? And were these technical replicates for the ELISA, or biological replicates?
- And how many replicates per MC for the immunogenicity experiments? Was each consortium tested with each of the samples from the 10 patients, or was the sample to limited for that? Assuming that each MC was tested in samples from each patient, was repeated measure ANOVA used to account for taking measures based on different treatments from the same samples?
- Some kind of summary figure, perhaps as supplement, summarizing what happened with each of the 504 strains would be useful. E.g. start with all 504 strains, then divide the strains that were resistant from those that weren't. Then divide the strains that were resistant based on peptidase activity, then based on inclusion in cytoplasm extracts and mock cultures to make it easy for a reader to figure out how many strains met which criteria.
- For Figure 4, I'm having trouble telling some of the bars apart, particularly MC7 and MC16. I also have no idea what you mean on line 392 by "Specimens in which gluten was digested by MC4 and MC16 showed the same (p>0.05)." Same as each other? Same as the negative control (but then how can you say this if p>0.05)? And is the p-value in line 389 for the ANOVA across all groups, or as part of the post-hoc testing? And saying "similar trend for IL-10 and IFN-gamma" is a bit frustrating, because I have no idea what the p-values are, so I don't know if you mean the same groups were significantly different or not.
Author Response
Dear reviewer, we would thank you for your suggestions. Please see the attachment.

Reviewer 2 Report
The authors have to be commended for this well-designed inquisitive study. The results are portrayed well and the manuscript is also written well. I have no major concerns. I would recommend the authors to include a brief discussion on exploring the feasibility (or challenges) of this work in animal models to further enrich the manuscript.
Author Response

(The authors gave the same response as above.)
